# Fast climate responses to emission reductions of aerosol and ozone precursors in China during 2013–2017

Jiyuan Gao[1], Yang Yang[1*], Hailong Wang[2], Pinya Wang[1], Huimin Li[1], Mengyun Li[1], Lili Ren[1], Xu Yue[1], Hong Liao[1]

[1]Jiangsu Key Laboratory of Atmospheric Environment Monitoring and Pollution Control, Jiangsu Collaborative Innovation Center of Atmospheric Environment and Equipment Technology, School of Environmental Science and Engineering, Nanjing University of Information Science and Technology, Nanjing, Jiangsu, China

[2]Atmospheric Sciences and Global Change Division, Pacific Northwest National Laboratory, Richland, Washington, USA

*Correspondence to yang.yang@nuist.edu.cn

**Abstract.** China has implemented a sequence of policies for clean air since year 2013 and the aerosol pollution has been substantially improved, but ozone (O$_3$) related issues arose. Here, fast climate responses to emission reductions in aerosols and O$_3$ precursors over China during 2013–2017 are investigated using the Community Earth System Model version 2 (CESM2). The overall decreases in aerosols produced an anomalous warming of 0.09 ± 0.10 °C in eastern China (22°N–40°N, 110°E–122.5°E), which is further intensified by the increase in O$_3$ in the lower troposphere, resulting in an enhanced warming of 0.16 ± 0.15 °C in eastern China. Reductions in industrial emissions contributed the most to the aerosol-induced warming, while emission reductions from residential sector induced a cooling effect due to a substantial decrease in light-absorbing black carbon aerosols. This implies that switching residential sector to cleaner energy is more effective to achieve climate and health co-benefits in China.

## 1 Introduction

Aerosol and tropospheric ozone (O$_3$) are two of the most critical air pollutants in the atmosphere, and have adverse effects on environment, human health, and ecosystems (Yang et al., 2014, 2017a). Due to increases in anthropogenic emissions associated with industrial development and economic growth (Zheng et al., 2018) and the intensification of unfavorable meteorological conditions (Yang et al., 2016), aerosol concentrations in China have dramatically escalated over the past several decades. To mitigate the serious air pollution, China issued the Air Pollution Prevention and Control Action Plan in 2013 (Clean Air Alliance of China, 2013), in which a decrease in PM$_{2.5}$ (particulate matter with diameter less than 2.5 μm) by 15%–25% by year 2017, compared to 2013, was proposed for various regions of China. The emissions of major air pollutants and precursors have been reduced since then and aerosol concentrations have substantially decreased across China (H. Li et al., 2021). In 74 key cities in China, the annual average of observed PM$_{2.5}$ concentrations decreased by 33.3% from 2013 to 2017 (Huang et al., 2018). However, as the aerosol decreases, surface O$_3$ pollution was getting worse, partly because the decrease in aerosols slowed down the sink of hydroperoxy radicals and thus stimulated O$_3$ production (Li et al., 2019). As also indicated from observations, the near-surface O$_3$ concentration increased by approximately 20% in China during 2013–2017 (Huang et al., 2018; Lu et al., 2018).

Both aerosols and O$_3$ play crucial roles in climate (Charlson et al., 1992; Chen et al., 2019; Koch et al., 2011; Li et al., 2016; Shindell et al., 2008; Xie et al., 2018; Yang et al., 2019, 2020). Through interacting with radiation and clouds, aerosols affect regional and global climate (Albrecht, 1989; Chen et al., 2010; Yang et al., 2017a, b). Effective radiative forcing (ERF) quantifies the energy gained or lost by the Earth system following imposed perturbation, and includes the instantaneous forcing plus adjustments from the atmosphere and surface (Smith et al., 2020). According to Forster et al. (2021), the total global aerosol ERF at the top of the atmosphere (TOA) estimated for 2019 relative to 1750 is –1.1 (–1.7 to –0.4) W m$^{-2}$, with –0.22 (–0.47 to 0.04) W m$^{-2}$ attributed to aerosol-radiation interactions and –0.84 (–1.45 to –0.25) W m$^{-2}$ from the aerosol-cloud interactions. O$_3$ has been recognized as one of the main contributors to radiative forcing, which exerts a global ERF of

0.47 (0.24 to 0.71) W m$^{-2}$. Tropospheric O$_3$ is a greenhouse gas and contributes the most to the O$_3$ ERF. According to their ERF, aerosols and O$_3$ changes from 1750–2019 induced a –0.50 (–0.22 to –0.96) °C cooling and a 0.23 (0.11 to 0.39) °C

warming, respectively, to the global surface air temperature (Forster et al., 2021).

Given that aerosol and O$_3$ are important short-lived climate forcers, a reduction in emissions of air pollutants for clean air always comes with climate consequences. The climate effects have been demonstrated in North America and Europe during the past decades when clean air actions were taken (Leibensperger et al., 2012a, 2012b; Turnock et al., 2015). Reductions in aerosol emissions in U.S. exerted a direct radiative forcing (DRF) by 0.8 W m$^{-2}$ and an indirect radiative

forcing (IRF) by 1.0 W m$^{-2}$ over eastern U.S., resulting in a 0.35 °C warming between 1980 and 2010 (Leibensperger et al., 2012a,b). Similarly, decreases in aerosols resulted in a DRF of 1.26 W m$^{-2}$ over Europe between 1980s and 2000s, and increases in O$_3$ exerted a radiative forcing of 0.05 W m$^{-2}$ in the meanwhile (Pozzoli et al., 2011). The clean air actions in Europe have been estimated to warm the surface air by 0.45 ± 0.11 °C between 1970 and 2010 (Turnock et al., 2015). Note that, the radiative forcing (RF) in these studies only includes the adjustment due to stratospheric temperature change, while

ERF consists all tropospheric and land surface adjustments and is commonly used recently.

The clean air actions in China have been reported to potentially affect radiative balance and regional climate in recent studies. Dang and Liao (2019) found that the reductions in aerosols led to a regional mean DRF of 1.18 W m$^{-2}$ over eastern China in 2017 relative to 2012 using the chemical transport model GEOS-Chem. Zheng et al. (2020) also reported that the decrease in aerosol emissions in China from 2006 to 2017 exerted an anomalous ERF of 0.48 ± 0.11 W m$^{-2}$ and further

caused a warming of 0.12 ± 0.02 °C in East Asia. Along with the decline of aerosols, O$_3$ concentrations also changed due to the clean air actions. The combined impacts of aerosol and O$_3$ changes on regional climate over China associated with clean air actions have not been studied. In addition, for the physical basis of climate policy decision making, it is valuable to know the relative roles of the sectoral sources contributing to the aerosol-induced climate change.

In this study, we examine the fast climate responses to emission reductions in air pollutants over China due to clean air

actions from 2013 to 2017, with the consideration of both aerosols and O$_3$ changes, using the Community Earth System Model Version 2 (CESM2) with its atmospheric component Community Atmosphere Model version 6 (CAM6). The climate impacts of aerosol emission reductions from individual sectors are also investigated through emission perturbation experiments.

## 2 Materials and methods

In this study, we perform simulations using the CAM6, the atmospheric component of CESM2, with a horizontal resolution of 0.9° latitude × 1.25° longitude and 32 vertical layers (Danabasoglu et al., 2020). In CAM6 major aerosol species, including sulfate (SO$_4^{2-}$), black carbon (BC), primary organic matter (POM), secondary organic aerosol (SOA), mineral dust, and sea salt, are represented by a modal aerosol scheme (Liu et al., 2016) with four lognormal modes (i.e., Aitken, accumulation, coarse, and primary carbon modes). PM$_{2.5}$ is calculated as the sum of SO$_4^{2-}$, BC, POM, and SOA in

this study. A comprehensive consideration of aerosol/$O_3$-radiation and aerosol-cloud interactions are included in the model. The Morrison-Gettelman cloud microphysics scheme version 2 (MG2, Gettelman and Morrison, 2015) is applied to forecast mass and number concentrations of rain and snow. The mixed phase ice nucleation depending on aerosols is also included (Hoose et al., 2010; Wang et al., 2014). Radiation transfer scheme uses Rapid Radiative Transfer Model for General circulation models (RRTMG, Iacono et al., 2008). Ozone mixing ratio is prescribed for use in radiative transfer calculations.

The ERF is decomposed into the forcing induced by aerosol-radiation interactions and aerosol-cloud interactions in this study based on the method proposed by Ghan et al. (2013) with an additional call to the radiation calculation.

Global three-dimensional tropospheric monthly $O_3$ concentrations below 450 hPa for years 2013 and 2017 are adopted from simulations using GEOS-Chem model v12.9.3, considering that it has a good performance in simulating $O_3$ concentration changes during 2013–2017 (K. Li et al., 2019, 2021). GEOS-Chem is a global model of atmospheric chemistry with fully coupled $O_3$–$NO_x$–hydrocarbon–aerosol chemical mechanisms, which has a horizontal resolution of 2° latitude ×

2.5° longitude and 47 vertical layers driven by the MERRA-2 (Modern-Era Retrospective analysis for Research and Applications Version 2) meteorological fields. The model simulations in 2013 and 2017 with one-year spin up use the same aerosol and precursor gas emissions as used in CAM6 and the results are interpolated to the same resolution as in CAM6. The details of the GEOS-Chem model simulations can be found in Li et al. (2022) and Yang et al. (2022). Note that, GEOS-

Chem model presents a strong decrease in $O_3$ concentrations in upper troposphere between 2013 and 2017, which is mainly attributed to the varying meteorological fields between 2013 and 2017. To minimize the impacts from the changes in meteorology, only $O_3$ data below 450 hPa from GEOS-Chem are used in CESM2 simulations, while keeping $O_3$ above 450 hPa unchanged, and are implemented by cycling the one-year data as monthly climatological mean.

Default anthropogenic and open biomass burning emissions of aerosols, aerosol precursors and $O_3$ precursors are

obtained from the CMIP6 (the Coupled Model Intercomparison Project Phase 6) (Hoesly et al., 2018; van Marle et al., 2017). Because CMIP6 emissions did not fully consider the emission reductions of clean air actions in China (Wang et al., 2021), anthropogenic emissions in China are replaced by the Multi-resolution Emission Inventory of China (MEIC) (Zheng et al., 2018) in both CESM2-CAM6 and GEOS-Chem simulations. The anthropogenic emission changes between 2013 and 2017 are shown in Fig. 1. Biogenic emissions are from the Model of Emissions of Gases and Aerosols from Nature version 2.1

(MEGAN v2.1, Guenther et al., 2012). All CAM6 experiments are forced by climatological mean sea surface temperatures (SSTs) and sea ice concentrations at year 2000 to characterize fast climate responses to the changes in air pollutants. Simulations are run for 20 years, with the last 15 years used in our analysis.

To investigate how climate was changed by aerosol and $O_3$ variations and sectoral contributions to aerosol-induced regional climate change during 2013–2017, the following experiments are conducted with CESM2-CAM6:

1. Base: global anthropogenic and natural emissions of aerosols and precursors and $O_3$ concentrations are fixed at year 2013.

2. AClean: same as Base, but anthropogenic emissions of aerosols and precursors over China are fixed at year 2017.

3. AClean_$O_3$: same as Base, but both anthropogenic emissions of aerosols and precursors and tropospheric $O_3$ concentrations over China are fixed at year 2017.

4. AClean_ENE/IND/RCO/TRA/SLV/WST/SHP: same as Base, but anthropogenic emissions of aerosols and their precursors from an individual sector, i.e., energy transformation and extraction (ENE), industrial combustion and processes (IND), residential, commercial and other (RCO), surface transportation (TRA), solvents (SLV), waste disposal and handling (WST) or international shipping (SHP) sectors, over China are fixed at year 2017.

The difference between Base and AClean is attributed to the impacts of aerosol emission reductions in China from 2013 to 2017, and the difference between Base and AClean_$O_3$ illustrates the combined effects of aerosol emission reductions and changes in tropospheric $O_3$ concentrations in China from 2013 to 2017. In addition, the comparison between Base and AClean_ENE/IND/RCO/TRA/SLV/WST/SHP quantifies the influences of aerosol emission reductions from the corresponding sector.

Hourly observations of PM$_{2.5}$ and $O_3$ concentrations across China in 2013 and 2017 derived from the China National Environmental Monitoring Centre (CNEMC) are applied to evaluate the model performance. The observations cover both urban and rural sites.

## 3 Changes in aerosols and $O_3$ in China from 2013 to 2017

To evaluate the model performance in simulating the changes in aerosol concentrations in China, Fig. 2a compares the 2013-to-2017 changes in annual mean surface concentrations of PM$_{2.5}$ between model simulations and ground measurements. The observed PM$_{2.5}$ reduced tremendously over eastern China, with maximum decreases exceeding 30 µg m$^{-3}$. CAM6 can reproduce the changes in spatial distribution of PM$_{2.5}$ concentrations, but strongly underestimates the magnitude of the concentration decreases with maximum decreases in the range of 12–18 µg m$^{-3}$. The low biases in CAM6 are caused by many factors including strong aerosol wet removal, uncertainties in new particle formation, coarse model resolution in global climate models, as well as the uncertainty of anthropogenic emissions of aerosols and precursor gases, which have been reported in many previous studies (Yang et al., 2017a, b; Zeng et al., 2021; Ren et al., 2021; Fan et al., 2018, 2022). The model generally captures the percentage decreases in PM$_{2.5}$ concentrations by about 20–40% in seven sub-regions over central-eastern China (Fig. 2c).

Accompanying the decreases in aerosols, annual mean near-surface $O_3$ concentrations increased in eastern China from 2013 to 2017. GEOS-Chem model catches the maximum increases of higher than 12 ppb (parts per billion) (15–30%) over the North China Plain (NCP) and Yangtze River Delta (YRD) (Figs. 2b and 2d), but underestimates the $O_3$ increases in other regions of China. The aerosol optical depth in central-eastern China also decreased from 2013 to 2017 in both model simulations and satellite retrievals, although the model still underestimates the decrease in satellite data (Fig. S1). In general, the models can reasonably reproduce the PM$_{2.5}$ decreases and $O_3$ increases in China, which give us the confidence in simulating climate impacts of emission reductions in air pollutants over China during 2013–2017. However, the larger

changes in $PM_{2.5}$ and $O_3$ in observations than in model simulations imply underestimated climate responses to the emission reductions in this study.

    Due to the implementation of clean air actions in China, $PM_{2.5}$ concentrations decreased dramatically between 25°N and 45°N averaged over 110–125.5°E in 2017 relative to 2013, and the $PM_{2.5}$ decline extended from the surface to about 850 hPa in the atmosphere (Fig. 3a). Table S1 summarizes the regional and seasonal mean simulated $PM_{2.5}$ column burdens in 2013

and 2017 over the seven sub-regions defined in Fig. 2. The annual mean column burden of $PM_{2.5}$ had the largest decrease over Sichuan Basin (SCB) by 15.2 mg m$^{-2}$ (30% relative to 2013) from 2013 to 2017, followed by 9.6 mg m$^{-2}$ (30%) over the Fenwei Plain (FWP) and 9.3 mg m$^{-2}$ (29%) over NCP. $PM_{2.5}$ burdens decreased by 6.5 mg m$^{-2}$ (23%) over YRD and by about 3–5 mg m$^{-2}$ (20%) over other sub-regions. The aerosol reductions in percentage did not show significant seasonal variations but similar values in all seasons over the seven sub-regions.

In contrast to the aerosol decreases, the simulated annual mean $O_3$ concentrations increased the most between 25°N and 45°N averaged over 110–125.5°E from 2013 to 2017 (Fig. 3b), partly because the reductions of aerosols can lead to a slowdown of the sink of $HO_2$ radicals in aerosol chemical processes and thus more radicals to accelerate the $O_3$ production (Li et al., 2019). Over eastern China, $O_3$ concentration increased from the surface to about 800 hPa. Meanwhile, $O_3$ concentrations decreased in the mid-troposphere in eastern China. It is because the mid-troposphere are relatively cleaner

than near the surface, which are the $NO_x$-limited regime, and $O_3$ concentrations decreased as $NO_x$ emissions decreased (Dufour et al., 2018). However, in the lower troposphere over eastern China, $O_3$ concentrations are limited by VOCs, which increased with reduced $NO_x$ emissions.

## 4 Fast climate responses to emission reductions in China

    As short-lived climate forcers, aerosols and $O_3$ exert considerable impacts on climate through perturbing the radiation

budget of the Earth. Along with the reductions in aerosol and precursor gas emissions due to clean air actions in China, the decreases in aerosol concentrations lead to an anomalous ERF of 1.18 ± 0.94 W m$^{-2}$ at TOA over eastern China in year 2017 relative to 2013 (Fig. 4a), which can potentially cause a regional warming effect. The anomalous ERF was largely induced by the aerosol-radiation interactions ($ERF_{ari}$, 0.79 ± 0.38 W m$^{-2}$) and the aerosol-cloud interactions also contributed to the ERF anomaly ($ERF_{aci}$, 0.44 ± 0.87 W m$^{-2}$) (Fig. 5). Note that due to the large uncertainties involved in the aerosol-cloud

interactions, changes in $ERF_{aci}$ and thus total aerosol ERF are not as statistically significant as $ERF_{ari}$.

    As a result of emission reductions in $O_3$ precursors, the $O_3$ concentrations increased in the lower troposphere and decreased in the mid-troposphere, resulting in a net ERF anomaly of 0.81 ± 0.92 W m$^{-2}$ at TOA over eastern China during 2013–2017 (Fig. 4b). There is an anomalous positive ERF anomaly over the Tibetan Plateau, which is due to the reduced surface albedo over this region (Fig. S2). The reduced surface albedo due to snow/ice melt over the Tibetan Plateau can

amplify the $O_3$-induced warming in China, even though the $O_3$ concentrations decreased over this particular region. The

positive ERF anomaly related to the near-surface $O_3$ increases enhanced the positive ERF produced by the aerosol decreases, leading to a total ERF anomaly of $1.99 \pm 1.25$ W m$^{-2}$ over eastern China (Fig. 4c).

Owing to the emission reductions, surface air temperature increased in China during 2013–2017, as the consequence of less solar radiation reflected to the space and more thermal radiation captured within the atmosphere. Over eastern China, surface air temperature increased by $0.09 \pm 0.10$ °C induced by anthropogenic aerosol emission reductions alone from 2013 to 2017 (Fig. 6a) and the intensified $O_3$ pollution exacerbated the temperature increase by $0.07 \pm 0.09$ °C in the meantime (Fig. 6b). The total aerosol and $O_3$ emission reductions from 2013 to 2017 induced a $0.16 \pm 0.15$ °C warming over eastern China, with statistically significant warming in the range of 0.3–0.5 °C between 30–40°N (Fig. 6c).

The regional surface air temperature changes over the seven sub-regions in China due to emission reductions of air pollutants are provided in Table 1. In FWP, temperature increased by $0.35 \pm 0.06$ °C between 2013 and 2017, equally attributed to the changes in aerosols and $O_3$. Temperature in NCP and SCB increased by $0.22 \pm 0.09$ °C and $0.26 \pm 0.08$ °C, largely attributed to changes in aerosols and $O_3$, respectively. Decreases in both aerosols and tropospheric $O_3$ above the surface caused a net surface cooling by $0.14 \pm 0.12$ °C in the Northeast Plain (NEP) in China. Note that, in this study we only focus on the fast climate responses over central-eastern China. Although temperature also increased or decreased in western China and outside China likely related to feedbacks or natural variability, there are few observational sites of air pollutants over these regions to verify the simulated pollutant changes and therefore large uncertainties exist in the simulated climate responses over these regions.

Although the air pollutants can influence precipitation through multiple microphysical and dynamical ways, the complicated aerosol-cloud interactions produced large uncertainties in the precipitation responses to the changes in air pollutants. Over eastern China, the reductions in emissions of air pollutants between 2013 and 2017 lead to the annual mean precipitation change by $-0.06 \pm 0.23$ mm day$^{-1}$ (Fig. S3). Neither the precipitation responses to changes in aerosols nor the $O_3$ are statistically significant at 90% confidence level over eastern China. In the simulations of this study, only fast climate responses are included with fixed SST at the climatological mean. Precipitation change is also driven by land-sea temperature differences over monsoon regions. Fixing SST in simulations can induce biases to the estimate of precipitation responses, which can be revisited using a fully coupled model configuration with both fast and slow climate responses included in future studies.

## 5 Impacts of aerosol reductions from individual sectors

To explore which emission sector contributed the most to the aerosol reduction-induced regional warming over eastern China, Fig. 7 shows the changes in column burden of PM$_{2.5}$, ERF$_{ari}$ and surface air temperature averaged in eastern China due to emission reductions of anthropogenic aerosols and precursors from individual sectors, and Table S2 summarizes the values. Among all the sectors, industrial emissions contributed the most to the column burden decrease of PM$_{2.5}$ in eastern China, accounting for 67% of the total burden decrease, followed by 27% due to emission reductions from the energy sector.

The $ERF_{ari}$ changes due to aerosol emission reductions in individual sectors from 2013 to 2017 are roughly in linear proportion to the burden changes but in the opposite direction. Reductions in aerosols from industrial and energy sectors exerted $ERF_{ari}$ anomalies of 0.50 W m$^{-2}$ (72% of the combined $ERF_{ari}$ anomaly from all sectors) and 0.20 W m$^{-2}$ (29%) and temperature anomalies of 0.063 and 0.025 °C, respectively, over eastern China. Declined surface transportation emissions introduced an $ERF_{ari}$ anomaly of 0.05 W m$^{-2}$ (8%) and a temperature anomaly of 0.007 °C, offset by the change in solvent usage. It is interesting that, different from most sectors, residential emissions reductions lead to a net cooling (–0.03 W m$^{-2}$ and –0.004 °C) in the context of the aerosol burden decreases over eastern China. It is because the residential heating sector releases a large amount of BC aerosol, which absorbs solar radiation and warms the atmosphere. With residential emissions reduced, decreases in BC resulted in less radiation trapped in the atmosphere and a negative $ERF_{ari}$ anomaly, although this effect was largely offset by the decreases in other scattering aerosols.

From 2013 to 2017, only about 10% of anthropogenic BC emission from residential sector was reduced in eastern China (Fig. S4). Previous studies have found that switching residential energy to cleaner energy prevented millions of premature deaths in China (Zhang et al., 2021). We suggest that the use of cleaner energy in the residential sector with less BC emissions is more effective to achieve climate and health co-benefits in China in the near future.

## 6 Conclusions and Discussions

Since year 2013, China has implemented a sequence of policies for clean air, which could have led to climate impacts through interactions between the changing air pollutants, radiation and clouds. In this study, the fast climate responses to emission reductions in air pollutants over China due to clean air actions from 2013 to 2017 are investigated based on CESM2-CAM6 simulations.

During 2013–2017, aerosol concentrations decreased significantly, whereas the simulated $O_3$ concentrations have an increase in the lower troposphere and a decrease in the mid-troposphere over eastern China. The aerosol decline produced an anomalous ERF of 1.18 ± 0.94 W m$^{-2}$ in eastern China, resulting in a 0.09 ± 0.10 °C warming during 2013–2017. An additional ERF of 0.81 ± 0.92 W m$^{-2}$ by the increases in $O_3$ in the lower troposphere enhanced the climate warming by 0.07 ± 0.09 °C, leading to an anomalous ERF of 1.99 ± 1.25 W m$^{-2}$ and a total 0.16 ± 0.15 °C warming in eastern China due to the changes in aerosols and $O_3$. It indicates that the recent growing $O_3$ pollution has strengthened the climate warming caused by aerosol emission reductions. Among all emission sectors, emission reductions in the industry sector contributed the most to the aerosol reduction-induced warming (72%), followed by the energy sector (29%). It is noteworthy that, associated with the reduced residential emissions, decreases in BC resulted in less solar radiation trapped in the atmosphere and caused a cooling effect, implying that switching residential sector to cleaner energy with less BC emissions is more effective to improve air quality and mitigate climate warming.

Different models have different climate responses to emission reductions due to uncertainties associated with the different physical, chemical and dynamical parameterizations and feedbacks. Table S3 compares the results in this study

with those from previous studies in the literature. Dang and Liao (2019) reported that reductions in aerosols from 2012 to 2017 had led to a regional mean DRF of 1.18 W m$^{-2}$ in eastern China using the GEOS-Chem model, which is higher than the ERF$_{ari}$ of 0.79 ± 0.38 W m$^{-2}$ in this study. They also showed a much weaker O$_3$ DRF of 0.08 W m$^{-2}$ than the ERF of 0.81 W m$^{-2}$ estimated here, which is probably because we only adopted tropospheric O$_3$ concentrations below 450 hPa from GEOS-Chem but they used total column O$_3$ for the radiation calculation with the influence of changing meteorological fields

between 2013 and 2017 included. With the coupled climate model CESM1, Zheng et al. (2020) found that emission reductions exerted a smaller ERF anomaly of 0.48 ± 0.11 W m$^{-2}$ and a stronger warming of 0.12 °C in East Asia during 2006–2017 compared to the 1.18 ± 0.94 W m$^{-2}$ and 0.09 ± 0.10 °C in this study averaged over eastern China during 2013–2017 considering fast climate responses alone.

As shown in this study, aerosol emission reductions in 2017, compared to 2013, led to a regional warming in China and
the increased tropospheric O$_3$ pollution further enhanced the warming, hindering climate warming mitigation goals. The connection between regional warming and emission reductions of air pollutants indicates the importance of a balance between air quality improvements and climate mitigations. Our results on sectoral contributions to climate impacts suggest that the residential sector is a good target for emission reduction to improve air quality and mitigate climate warming simultaneously yet reducing aerosol emissions in other sectors, especially the industry sector, is likely to accelerate the
regional warming in China.

There are some limitations and uncertainties in the study. Firstly, only fast climate responses are considered in our study, while the emission reductions could also influence climate response through slow oceanic processes and air-sea interactions, which can be improved by conducting fully coupled atmosphere-ocean simulations in future studies. Samset et al. (2016) showed that the fast precipitation response to changes in aerosols dominated the slow oceanic response over land
of East Asia. However, to what extent the fast processes contributed to the temperature response needs further study. Neglecting the slow climate response here could lead to an incomplete aerosol climate effect. Secondly, the model significantly underestimates the PM$_{2.5}$ decrease in China during 2013–2017, which is caused by many factors including strong aerosol wet removal, uncertainties in new particle formation, the coarse model resolution, and the uncertainty in anthropogenic emissions of aerosols and precursor gases (Yang et al., 2017a, b; Zeng et al., 2021; Ren et al., 2021; Fan et al.,
2022, 2018). The low bias in estimated aerosol decreases may result in an underestimation of the simulated climate responses in CAM6. Thirdly, nitrate and ammonium aerosols, which are not treated in current version of CESM2, also changed from 2013 to 2017 (Xu et al., 2019) and should have impacted on climate, although nitrate concentration in Beijing changed slightly during this time (Zhang et al., 2020). Fourthly, only 20-year simulations were performed in this study, longer simulations with ensemble members may present a more robust result. Finally, only one model is used in our study, a
potential model dependence of climate responses to aerosol reductions needs further investigation using multi-model ensemble simulations. Furthermore, several interesting issues can be investigated in the future. For example, our results only illustrate the impacts of China's emission changes of air pollutants on China's regional climate, but regional climate changes

in China can respond to emission changes outside China, e.g., South Asia, and remote climate responses to China's emission reductions deserve further studies as well.


*Acknowledgments*

This study was supported by the (grant 2019YFA0606800 and 2020YFA0607803), the National Natural Science Foundation of China (grant 41975159) and Jiangsu Science Fund for Distinguished Young Scholars (grant BK20211541). H.W. acknowledges the support by the U.S. Department of Energy (DOE), Office of Science, Office of Biological and
Environmental Research (BER), as part of the Earth and Environmental System Modeling program. The Pacific Northwest National Laboratory (PNNL) is operated for DOE by the Battelle Memorial Institute under contract DE-AC05-76RLO1830.

*Data availability*

Observed $PM_{2.5}$ and $O_3$ concentrations are available at https://doi.org/10.5281/zenodo.5833003 (last access: April 2022). The GEOS-Chem model is available at https://zenodo.org/record/3974569# (last access: April 2022). The CESM2 model is available at https://www.cesm.ucar.edu/models/cesm2/release_download.html (last access: April 2022). The MEIC inventory can be downloaded at http://meicmodel.org/?page_id=541&lang=en (last access: April 2022). Our model results
are available at https://doi.org/10.5281/zenodo.6418003 (last access: April 2022).

*Competing interests*
The authors declare that they have no conflict of interest.

*Author contribution*
YY designed the research; JG performed the model simulations and analyzed the data. All authors discussed the results and wrote the paper.

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

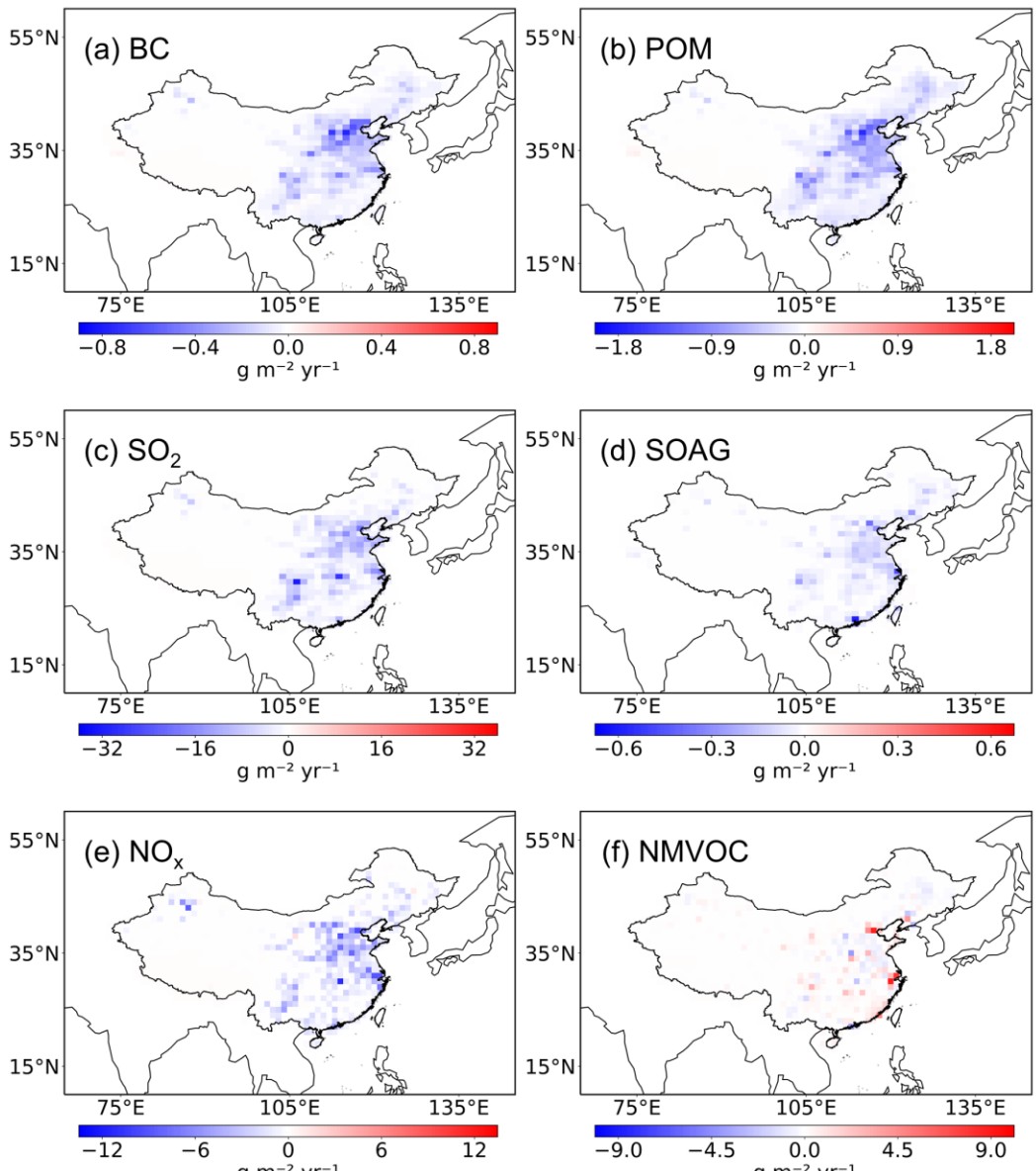


**Figure 1.** Spatial distributions of emission differences of aerosol, aerosol precursors and ozone precursors including black carbon (BC), primary organic matter (POM), sulfur dioxide (SO₂), secondary organic aerosol gas (SOAG), nitrogen oxides (NO$_x$) and non-methane volatile organic compounds (NMVOC) between 2013 and 2017 (2017 minus 2013). The anthropogenic emission data are derived from MEIC.

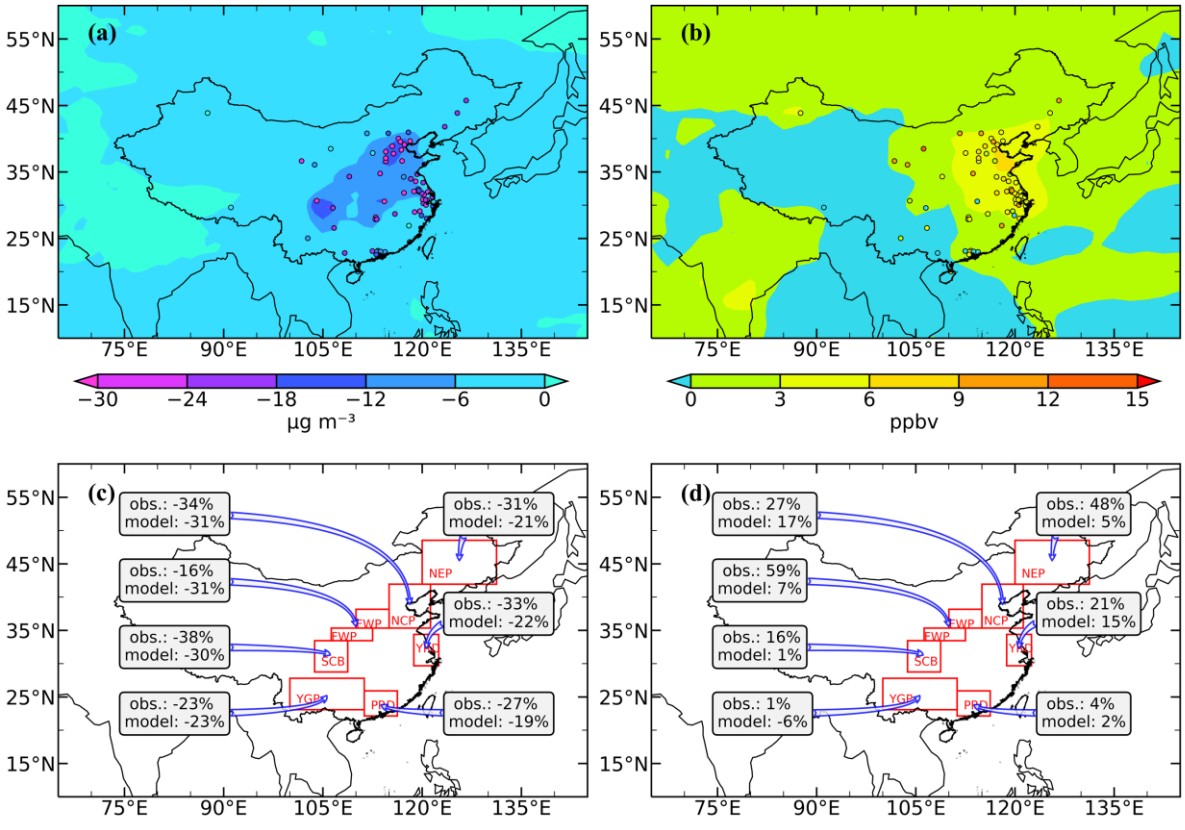


**Figure 2.** Spatial distributions of differences in observed (obs., circles) and simulated (model, shades) annual mean near-surface (a) PM$_{2.5}$ (µg m$^{-3}$) and (b) O$_3$ (ppbv) concentrations over China between 2013 and 2017 (2017–2013) and (c, d) the percentage changes averaged over seven sub-regions of China, including the North China Plain (NCP, 35°N–41°N, 114°E–

120°E), Sichuan Basin (SCB, 28°N–33°N, 103°E–108°E), Yangtze River Delta (YRD, 29°N–34°N, 118°E–121.5°E), Pearl River Delta (PRD, 21.5°N–25°N, 111°E–116°E), Northeast Plain (NEP, 41°N–48°N, 120°E–130°E), the Yunnan–Guizhou Plateau (YGP, 23°N–27°N, 100°E–110°E), and the Fenwei Plain (FWP, 33°N–35°N, 106°E–112°E and 35°N–38°N, 110°E–114°E). The modelled changes in PM$_{2.5}$ and O$_3$ are the differences between Base and AClean simulations (AClean–Base) and between AClean and AClean_O$_3$ (AClean_O$_3$–AClean), respectively. Modelled PM$_{2.5}$ data are from CESM2 simulations

in a/c and modelled O$_3$ data are from GEOS-Chem simulations in b/d.

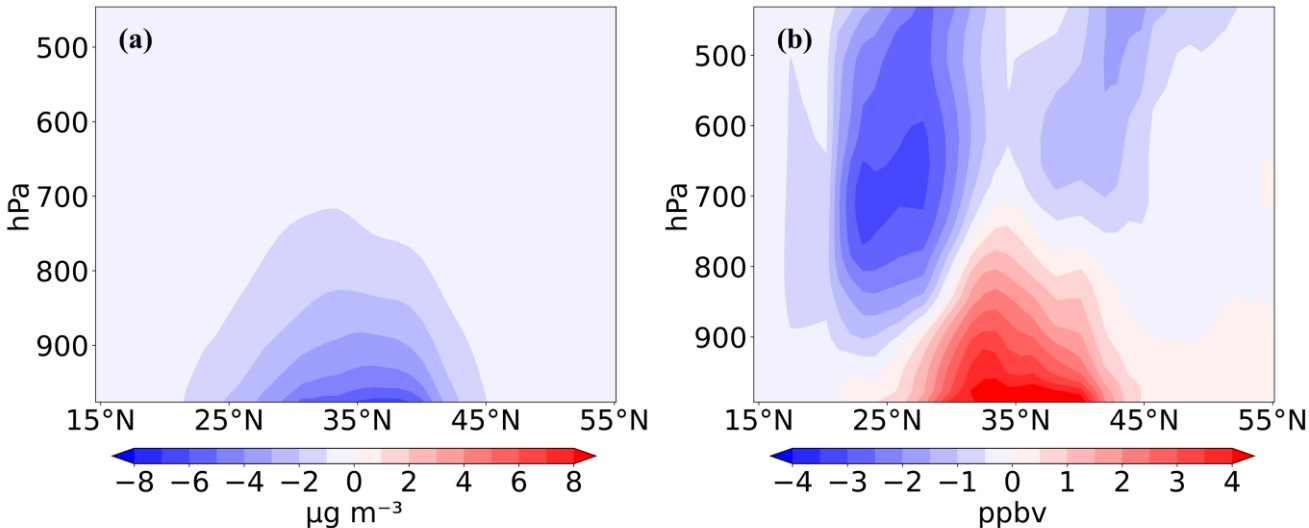

**Figure 3.** Pressure–latitude cross-section averaged over 110–125.5°E for differences in simulated annual mean (a) $PM_{2.5}$ (μg m$^{-3}$) and (b) $O_3$ (ppbv) concentrations due to emission reductions of aerosol pollutants between 2013 and 2017 (2017 minus 2013). Modelled $PM_{2.5}$ data are from CESM2 simulations and modelled $O_3$ data are from GEOS-Chem simulations.

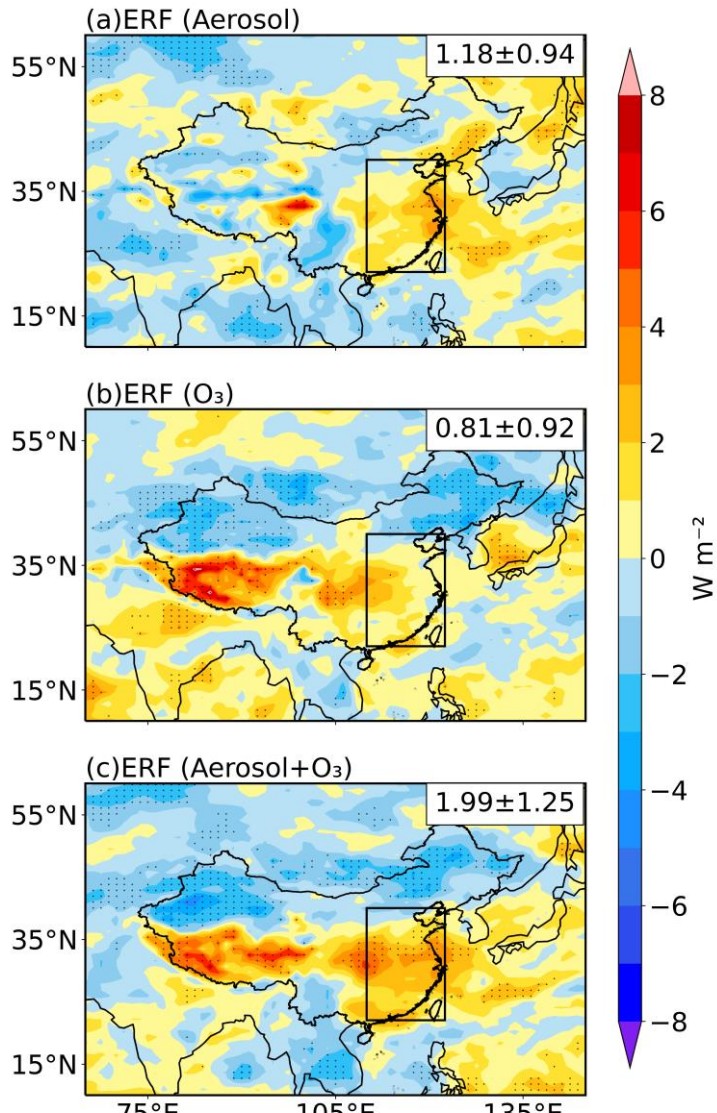


**Figure 4.** Spatial distributions of changes in annual mean effective radiative forcing (ERF, W m$^{-2}$) of (a) aerosols, (b) tropospheric O$_3$, and (c) both aerosols and tropospheric O$_3$ in 2017 relative to 2013. ERF of aerosols, O$_3$ and both aerosols and O$_3$ are calculated as the differences in net radiative fluxes at the top of the atmosphere between Base and AClean
simulations (AClean–Base), between AClean and AClean_O$_3$ (AClean_O$_3$–AClean), and between Base and AClean_O$_3$ (AClean_O$_3$–Base), respectively. Differences in areas that are statistically significant at 90 % from a two-tailed *t* test are stippled. Regional average and standard deviation of the change in eastern China (22°N–40°N, 110°E–122.5°E) are noted at the upper-right corner of each panel.

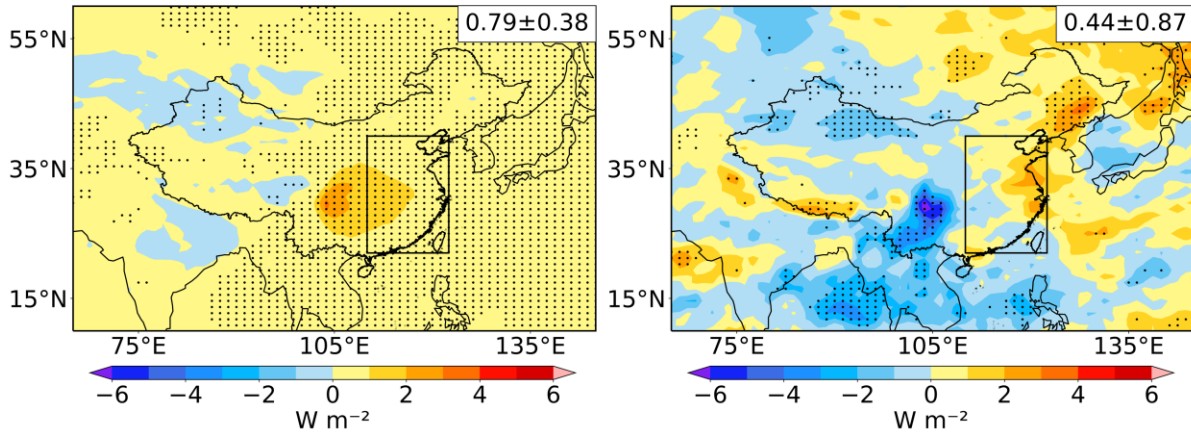

**Figure 5.** Spatial distributions of ERF$_{ari}$ (effective radiative forcing induced by aerosol-radiation interactions) (left) and ERF$_{aci}$ (effective radiative forcing induced by aerosol-cloud interactions) (right) differences between Base and AClean. The regional and annual mean difference in eastern China (22°N–40°N, 110°E–122.5°E) are indicated at the upper-right corner of each panel. Differences in areas that are statistically significant at 90 % from a two-tailed *t* test are stippled.

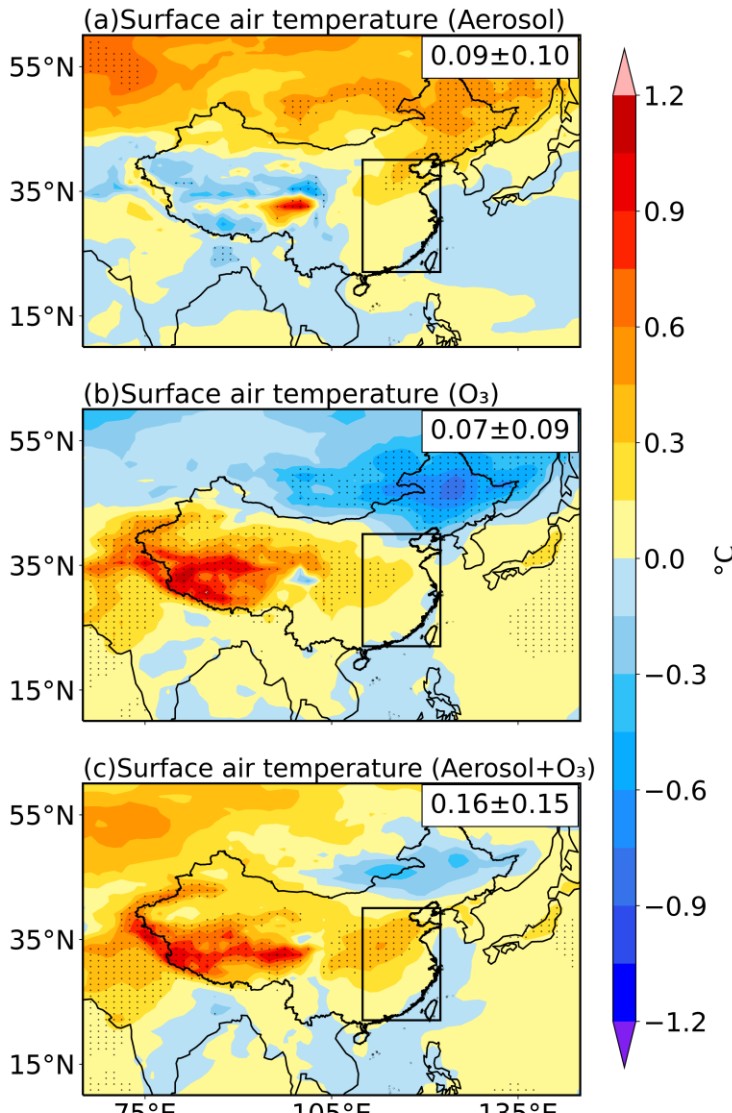

**Figure 6.** Spatial distributions of differences in surface air temperature (°C) due to the changes in (a) aerosols, (b) O₃, and (c) both aerosols and O₃ between 2013 and 2017, calculated as the differences between Base and AClean simulations (AClean–Base), between AClean and AClean_O₃ (AClean_O₃–AClean), and between Base and AClean_O₃ (AClean_O₃–Base), respectively. Differences in areas that are statistically significant at 90 % from a two-tailed *t* test are stippled. Regional average and standard deviation of the change in eastern China (22°N–40°N, 110°E–122.5°E) are noted at the upper-right corner of each panel.

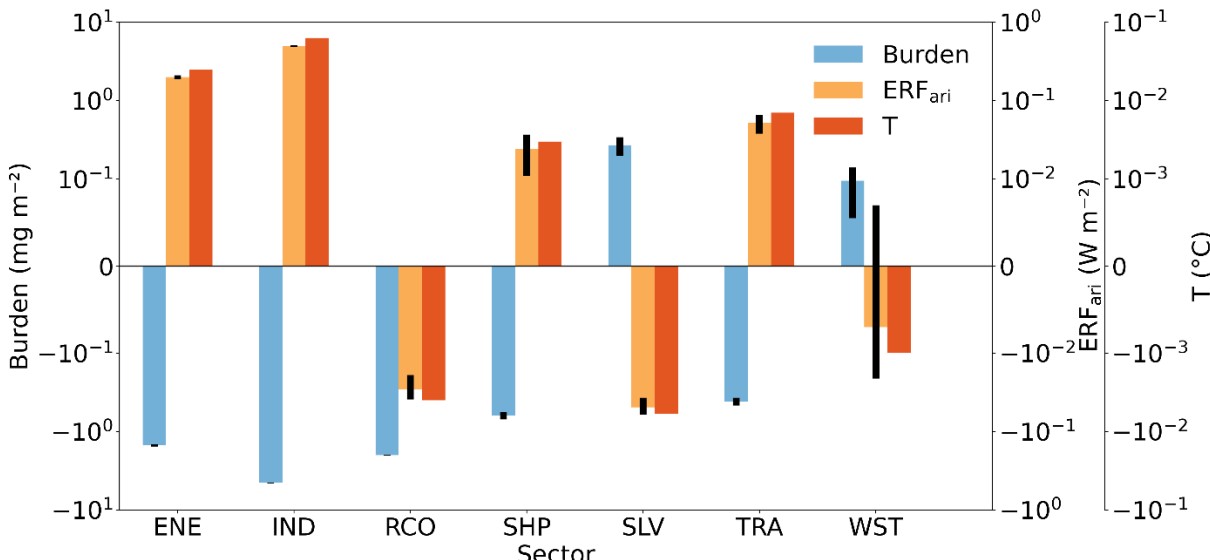

**Figure 7.** Changes in aerosol column burden, effective radiative forcing due to aerosol-radiation interactions (ERF$_{ari}$) and surface air temperature (T) in 2017 relative to 2013 averaged over eastern China due to emission reductions of aerosols and precursors from individual sectors, including energy transformation and extraction (ENE), industrial combustion and processes (IND), residential, commercial and other (RCO), surface transportation (TRA), solvents (SLV), waste disposal and handling (WST) and international shipping (SHP). Error bars of burden and ERF$_{ari}$ indicate 1σ. Note that scales are logarithmic.

**Table 1**. Regional and seasonal mean ERF (effective radiative forcing) (W m$^{-2}$) and surface air temperature (°C) changes induced by aerosol and/or O$_3$ changes between 2013 and 2017.

| Region | Pollutant | ERF (W m$^{-2}$) | Surface air temperature (°C) |
|---|---|---|---|
| NCP | Aerosol | 0.68±0.76 | 0.26±0.09 |
| | O$_3$ | -0.41±0.97 | -0.05±0.15 |
| | Aerosol+O$_3$ | 0.27±1.18 | 0.22±0.14 |
| SCB | Aerosol | -0.94±1.57 | 0.01±0.09 |
| | O$_3$ | 3.08±0.78 | 0.25±0.04 |
| | Aerosol+O$_3$ | 2.13±1.19 | 0.26±0.08 |
| YRD | Aerosol | 2.74±0.57 | 0.05±0.03 |
| | O$_3$ | 0.31±0.40 | 0.04±0.04 |
| | Aerosol+O$_3$ | 3.05±0.47 | 0.09±0.06 |
| PRD | Aerosol | 1.20±0.52 | 0.05±0.02 |
| | O$_3$ | 0.97±0.74 | 0.01±0.02 |
| | Aerosol+O$_3$ | 2.17±0.58 | 0.06±0.03 |
| NEP | Aerosol | 0.65±1.13 | 0.41±0.09 |
| | O$_3$ | -2.21±0.81 | -0.56±0.19 |
| | Aerosol+O$_3$ | -1.55±0.89 | -0.14±0.12 |
| YGP | Aerosol | -0.68±1.38 | -0.02±0.05 |
| | O$_3$ | 0.62±0.63 | 0.06±0.03 |
| | Aerosol+O$_3$ | -0.05±1.50 | 0.05±0.05 |
| FWP | Aerosol | 0.75±0.38 | 0.17±0.06 |
| | O$_3$ | 1.34±0.97 | 0.18±0.07 |
| | Aerosol+O$_3$ | 2.09±1.20 | 0.35±0.06 |

580