# Peer review of "Fast climate responses to emission reductions of aerosol and ozone precursors in China during 2013–2017"

_Atmospheric Chemistry and Physics, 2022_

## Referee Comment (RC2)

The manuscript by Gao et al. studies the climate responses to emission reductions in air pollutants over China due to clean air actions from 2013 to 2017, investigates both aerosols and ozone changes and their climate impacts by conducting several experiments using CESM2 model. The topic has wide implication for emission reduction policy decision making over China and fits the scope of the general ACP readership. This paper is overall well written, but there are several issues need to be addressed before the manuscript can be accepted for publication.

Major:

1. The model results significantly underestimate the PM2.5 decrease compared with observation (Fig.2), which contributes to the uncertainty of this study. It would be interesting to quantify to what extent the model bias influences the estimated climate impacts.
2. The authors investigated the climate response by conducting simulations with fixed SST at the climatological mean. I wonder how much does the slow and fast response contribute to the total climate response respectively? Though the authors stated that they will revisit this issue using a fully coupled model configuration with both fast and slow climate responses included in future studies, it is suggested to discuss the uncertainties due to neglecting the slow climate response in this paper.
3. I would suggest the authors to provide an in-depth discussion in the discussion section on these uncertainties, including the model bias, the neglect of slow response, the neglect of nitrate and ammonium, etc. It is better to have error bars on the simulated results or at least discuss the possible bias ranges. In addition, as stated in L215, different chemical and physical schemes contribute some uncertainties, leading to the differences compared to previous studies. Thus it is better to list the specific parameterizations of different models in Table S3.

Minor:

1. L79, 'A comprehensive consideration of aerosol/O3-radiation and aerosol-cloud interactions are included in the model.' How are these processes considered specifically in the model? I suggest authors to introduce these schemes in detail, or at least show some references.
2. L83, it is better to list some reference about ozone simulation in GEOS-Chem here.
3. In section 2, please add some introductions about observations used in this study.
4. L133, change 'other sub-regions' to 'over other sub-regions'.
5. L136-L137, better to list some references here.
6. Figure 1, better to mention the MEIC inventory in figure caption.
7. Figure S1, the color bar is not shown.

---

## Author Comment (AC1)

**Overall comments:**

I find the manuscript by Gao et al. interesting and fairly well-written. China's emissions of aerosol and ozone precursors declined after 2013 due to measures to improve air quality, and studies of the climate effects of these emission reductions are clearly of interest. While the topic has been investigated before, Gao et al. applies a state-of-the-art climate model (CESM2) for their quantifications. However, I have concerns regarding the methodological setup of the study and have several questions and comments that need to be addressed. My main concerns are:

We thank the reviewer for the constructive suggestions, which are very helpful for improving the clarity and reliability of the manuscript. Please see our point-by-point responses (in blue) to your comments below.

The study uses prescribed sea-surface temperature simulations, which is commonly used to quantify effective radiative forcing, but it is not ideal for characterizing climate responses, such as surface temperature change. While the authors state that they characterize fast climate responses, this is not clear enough in the manuscript – e.g., reading the current title and abstract gives the impression that the total climate response is studied. The fast temperature response is only part of the total response (which involves changes in sea-surface temperatures) but quantifying this requires coupled atmosphere-ocean simulations. Unless the authors want to carry out fully coupled simulations, my suggestion is to concentrate more on the ERF results and de-emphasize or remove the results showing surface temperature changes.

Thank you for your suggestion. Fast climate responses to aerosol changes are of great importance to climate change, and the temperature changes due to the fast climate responses to aerosols have been examined in many previous studies (e.g., Liu et al., 2018; Samset et al., 2016; Yang et al., 2020; Zanis et al., 2020). However, we also agree that this information was not clearly given in the manuscript. To clarify this information, we have changed the title to "Fast climate responses to emission reductions of aerosol and ozone precursors in China during 2013–2017". Besides, the main text has been revised to emphasize the focus of fast climate responses.

The choice of using GEOS-Chem to calculate ozone makes the results less consistent and seems a bit odd given that CESM2 has a detailed tropospheric chemistry package (Emmons et al., 2020). Nevertheless, details about the GEOS-Chem simulations, and clarifications of how the GEOS-Chem results are used in CESM2, are needed, especially given the factor 10 difference in the ERF due to ozone changes in this study compared to an earlier study using GEOS-Chem.

The tropospheric chemistry in CESM2 is an updated MOZART mechanism. We have tested the trends in surface $O_3$ concentrations in China in CESM1 with MOZART mechanism. The simulated $O_3$ concentrations showed a decreasing trend in the recent decade, in opposite to the observations. That is why we used GEOS-Chem model instead to archive $O_3$ data, which showed a good performance in simulating $O_3$ concentration changes in China during 2013–2017 (Li et al., 2019a, b, 2021).

Details about the GEOS-chem simulations and clarifications of how the GEOS-Chem results are used in CESM2 are as followed and have been added in section 2:

Global three-dimensional tropospheric monthly $O_3$ concentrations for years 2013 and 2017 are adopted from simulations using GEOS-Chem model v12.9.3, considering that GEOS-Chem has a good

performance in simulating ozone concentration changes during 2013–2017 (Li et al., 2019a, b, 2021). It is a global model of atmospheric chemistry with fully coupled $O_3$–$NO_x$–hydrocarbon–aerosol chemical mechanisms, which has a horizontal resolution of 2° latitude × 2.5° longitude and 47 vertical layers driven by the MERRA-2 (Modern-Era Retrospective analysis for Research and Applications Version 2) meteorological fields. The model simulations in 2013 and 2017 with one-year spin up use the same aerosol and precursor gas emissions as used in CAM6 and the results are interpolated to the same resolution used in CAM6. The details of the GEOS-Chem model simulations can be found in Li et al. (2022) and Yang et al. (2022).

The possible reason why there exists a large difference in the ERF due to ozone changes between Dang and Liao (2019) and our study is that we only adopted tropospheric $O_3$ concentrations below 450 hPa from GEOS-Chem while they used total column $O_3$ for the radiation calculation. The ozone decreases in the upper troposphere and the stratosphere related to changes in meteorology would offset some radiative effects in Dang and Liao (2019), making their results smaller than our results. We have discussed this difference in the manuscript.

More work is needed to show that the CESM2 simulations realistically reproduce observed aerosols and the aerosol decline between 2013 and 2017. In particular, the natural aerosols that are modelled in CESM2 by default (dust and sea salt) should be included in the calculation of PM2.5.

[Figure]

Figure A. Spatial distributions of differences in observed (obs., circles) and simulated (model, shades) annual mean near-surface PM2.5 (µg m$^{-3}$) concentrations over China between 2013 and 2017 (2017–2013) without (left) and with (right) dust and sea salt aerosols.

In this study, we only focus on the fast climate responses to changes in anthropogenic emissions. We have tested the comparison by adding dust and sea salt in the calculation of PM2.5. However, as the figure shows above, the pattern of PM2.5 decreases changed little after adding dust and sea salt because we only cut anthropogenic emissions in the simulations. Natural emissions are online calculated and only affected by aerosol-induced changes in meteorological fields, which are unlikely to have a large impact on total aerosol variation.

**Specific comments:**

Title: Given that the full climate response is not investigated in these prescribed SST simulations, the first part of the title ("Climate impacts of") should be changed to "Radiative forcing due to" or similar. The second part of the title ("emission reductions") should also be changed by specifying that this only applies to aerosol and ozone precursors. The present title implies that the full climate response to all emissions (including long-lived greenhouse gases) is investigated, and this is not the case.

We have changed the title to "Fast climate responses to emission reductions of aerosol and ozone precursors in China during 2013–2017".

L21-25: I do not think "accelerated" is the right word here (enhanced or increased would be more correct). In any case, the surface temperature changes given here do not include the slow temperature response, only the fast responses over land, and are therefore most probably underestimated. It would therefore be better to change the focus from "climate responses" to "radiative forcing" and give the ERF numbers rather than the temperature changes. Similar modifications could be done elsewhere in the manuscript.

The word "accelerated" has been revised to "enhanced". "Fast climate response" has been emphasized in the abstract. Temperature changes due to the fast climate responses to aerosols have been examined in many previous studies (e.g., Yang et al., 2020; Zanis et al., 2020), which are also of great importance to climate change.

L27-28: Before making this conclusion, I think an investigation of the potential for emission reductions in this sector is needed – please see my comment further down (L227-230).

Please see our response below.

L32: There are two papers by Yang et al. (2017) in the reference list. Please specify which one.

Specified.

L37-38: Better to say "emissions of major air pollutants and precursors"?

Thanks for your suggestion. Revised.

L47: Rather than citing the whole report, it is better to cite the actual chapter, in this case Forster et al. (2021) (Ch. 7). It is then much easier for the reader to find the relevant information.

Revised.

L47-48: Effective radiative forcing is such a central term here and I think it should be defined/explained briefly.

We have now defined effective radiative forcing in the main text: "Effective radiative forcing (ERF) quantifies the energy gained or lost by the Earth system following imposed perturbation, which includes the instantaneous forcing plus adjustments from the atmosphere and surface (Smith et al., 2020)."

L49: The ERF_ari uncertainty range is -0.47 to 0.04 W m-2 (i.e., not a minus sign in front of 0.04) – see Table 7.8 in IPCC AR6.

Corrected.

L50-51: Should mention that tropospheric ozone is a greenhouse gas and contributes the most to the ozone ERF.

Added as suggested.

L56-65: Can you explain very briefly for the reader how the direct and indirect radiative forcing differ from ERF, so that it is easier to compare the numbers between the studies?

We have now explained as "Note that, the radiative forcing (RF) in these studies only includes the adjustment due to stratospheric temperature change, while ERF consists all tropospheric and land surface adjustments and is commonly used recently."

Section 2: There is no description or reference to the surface measurements of PM2.5 and O3. How were the measurements performed and are they representative of urban conditions? Or rural/background?

We have now added the content in section 2: "Hourly observations of $PM_{2.5}$ and $O_3$ concentrations over China in 2013 and 2017 derived from the China National Environmental Monitoring Centre (CNEMC) are applied to evaluate the model performance." It is a national observational network in China established in 2013 and the data were quality controlled and widespread evaluated and used in many previous studies.

L78-79: Why are dust and sea salt not included in the calculation of PM2.5? These are standard output in CESM2 and I suppose they could make a substantial contribution to PM2.5 levels.

Yes, the natural aerosols largely contribute to $PM_{2.5}$. However, we focused on the changes in anthropogenic emissions. The changes in $PM_{2.5}$ are similar with and without natural aerosols in the $PM_{2.5}$ calculation, as we responded above.

L80-84: The CESM2 gas chemistry package (Emmons et al., 2020) could easily have been included and would make the study much more consistent because of two-way interaction between gases and aerosols, and because the same meteorology would have been used for calculating both gases and aerosols. Why was GEOS-Chem used instead? Was it because of computational requirements? There is no reference to the GEOS-Chem model, and more information about these simulations is needed. For example, what aerosol compounds are included in these simulations? How long were the simulations? What resolution is used?

Please see our response above.

L87-89 / Figure 1: It would have been very useful to see a plot (e.g., in the supplementary) of the time evolution of emissions, for instance from 2000-2019, to see how 2013 and 2017 compare to the other years. Comparison with the newest version of the CEDS emission inventory

(https://github.com/JGCRI/CEDS), which better accounts for emission reductions in China (in contrast to the CEDS version used in CMIP6), can also be considered.

The time evolution of emissions in China from 2010–2017 from MEIC inventory has been given in Zheng et al. (2018), as shown below. Many studies have revealed that the CMIP6 emissions did not fully consider the emission reductions in China during 2013–2017 (Cheng et al., 2021; Wang et al., 2021). Although the latest CEDS v_2021_04_21 updated to 2019 did consider the emission changes in China, the comparison between CEDS v_2021_04_21 and MEIC to find which is the better inventory is out of the scope of this study.

[Figure]

Figure B. China's anthropogenic emissions by sector and year. The species plotted here include (a) $SO_2$, (b) $NO_x$, (c) NMVOCs, (d) $NH_3$, (e) CO, (f) TSP, (g) $PM_{10}$, (h) $PM_{2.5}$, (i) BC, (j) OC, and (k) $CO_2$. Chinese emissions are divided into six source sectors (stacked column chart): power, industry, residential, transportation, agriculture, and solvent use. Besides the actual emissions data, two emission scenarios are presented to provide emission trajectories when assuming activity (inverted triangle) or pollution control (upright triangle) frozen at 2010 levels. This figure is from Zheng et al. (2018).

L90: I assume the biogenic emissions are only included in the GEOS-Chem simulations?

Biogenic emissions are also included in CESM2 for the simulation of secondary organic aerosol (SOA).

L92: "present-day level" - please specify which year(s)

It refers to year 2000. Specified in the manuscript.

L93: Given the weak statistical significance in most of the results, are 20-year simulations long enough? For comparison, Zheng et al. (2020) ran the CESM1 model (fixed SST) for 60 years (analyzing the last 40 years) and got quite robust ERF numbers.

We agree that longer simulations may get a more robust result. However, the CESM2-CAM6 is much more expensive than CESM-CAM5 used in Zheng et al. (2020). With fixed sea surface temperature (SST), the 20-year simulations should be sufficient for the fast climate response analysis. Many studies also conducted relative short simulations less than 20 years with fixed SST to examine the fast climate responses to aerosol forcings (e.g., Liu et al., 2018).

L115-119:

- "lack of nitrate and ammonium representation": I agree that the lack of ammonium nitrate leads to smaller PM2.5 values, but that does not necessarily mean that it contributes to the underestimation of the decrease in PM2.5. Are there any indications in the ammonia (NH3) emission data that there has been a decrease between 2013 and 2017? Ammonium nitrate formation is heavily dependent on the levels of sulphate, and the strong decrease in SO2 emissions implies that ammonium nitrate concentrations would actually increase.

  Thank you for the comment. Although nitrate mass fraction increased, nitrate concentration changed slightly during the time (Zhang et al., 2020). Therefore, the lack of nitrate and ammonium representation would not result in the underestimate $PM_{2.5}$ concentration decline. We have removed this explanation in the text.

- "absence of natural aerosols in the calculation of modeled PM2.5": I do not think that the lack of natural aerosols would influence the underestimated decrease in PM2.5? But including the natural aerosols would clearly give more realistic absolute concentration values (not changes).

  Agree. Removed now.

L118-119: Since the model captures the relative differences better than the absolute differences, it could perhaps indicate that the actual PM2.5 concentrations are underestimated in the model. It would be useful to compare the model against observations of PM2.5 concentrations for 2013 and 2017 separately, for instance as a supplementary figure (similar to Fig. 2a). Underestimation of PM2.5 could also be partly caused by coarse model resolution, which I assume is 0.9x1.25 degrees (this should be stated in methods)? Are the observations primarily from urban areas? If so, the model is not expected to reproduce these observations, and one method that could be used to account for this is the so-called urban increment factor (see e.g., Aunan et al., 2018).

[Figure]

Figure C. Simulated annual mean near-surface PM$_{2.5}$ concentrations in 2013 (left) and 2017 (right).

The observed data from CNEMC covers both urban and rural sites. According to the Fig. C above, the model underestimates PM$_{2.5}$ concentrations in both 2013 and 2017. The low biases in CAM6 are caused by many factors including strong aerosol wet removal, uncertainties in new particle formation, the coarse model resolution, and the uncertainty in anthropogenic emissions of aerosols and precursor gases, which have been reported in many previous studies (Yang et al., 2017a, b; Zeng et al., 2021; Ren et al., 2021; Fan et al., 2018, 2022). Not only CESM2, many climate models have large low biases in simulating aerosol concentrations over China, which requires in-depth analysis and effort to solve it in future studies.

L123-124: I am not totally convinced. In terms of simulating climate impacts of aerosol reductions, a useful comparison would be aerosol optical depth (AOD) from the model (should be standard output) against satellite observations (MODIS data available from https://giovanni.gsfc.nasa.gov/).

[Figure]

Figure S1. Spatial distributions of annual mean aerosol optical depth (AOD) differences from CESM2 simulations (a) and MODIS (Moderate Resolution Imaging Spectroradiometer, b) over China between 2013 and 2017 (2017–2013)

We have added Figure S1 in the supplementary material to show that the aerosol optical depth in central-eastern China also decreased from 2013 to 2017 in both model simulations and satellite retrievals, although the model still underestimates the decrease in satellite data (Fig. S1).

L147-149: How was the ERF separated into aerosol-radiation interactions and aerosol-cloud interactions? Did you apply double radiation calls?

The ERF is decomposed into the forcing induced by aerosol-radiation interactions and aerosol-cloud interactions in this study based on the method proposed by Ghan et al. (2013) with an additional call to the radiation calculation. We have added the text in the Materials and methods.

L149-153: Again, I think the statistics would have been better with more years and/or ensemble members.

We have also added this limitation in the last paragraph as "Fourthly, only 20-year simulations were performed in this study, longer simulations with ensemble members may present a more robust result."

L151-154 / Figure 4b: There is a strong positive increase in ERF due to O3 changes over the Tibetan Plateau, despite decrease in near-surface O3 in this region (Fig. 2b) and decreasing O3 at height (Fig. 3b). Any reason why?

[Figure]

Figure S2. Spatial distributions of differences in surface albedo the changes in O3 between 2013 and 2017, calculated as the differences between AClean and AClean_O3 (AClean_O3–AClean. Differences in areas that are statistically significant at 90 % from a two-tailed t test are stippled.

There is an anomalous positive ERF anomaly over the Tibetan Plateau, which is due to the reduced surface albedo over this region (Fig. S2). The reduced surface albedo due to snow/ice melt over the Tibetan Plateau can amplify the $O_3$-induced warming in China, even though the $O_3$ concentrations decreased over this particular region. We have added the explanation in the manuscript.

Figure 7 and Table S2: It would be good to see uncertainties for these numbers.

Added for Figure 7 and Table S2.

L215-218: The Table S3 caption states that Dang and Liao (2019) considered ERF_ari, while the text gives the number as the direct effect. ERF_ari includes also semi-direct effects in addition to the direct effect (see e.g., Fig. 7.3 in Boucher et al., 2013) and semi-direct effects are particularly important for BC. Are semi-direct effects included in the number from Dang and Liao (2019)?

Thank you for your reminding. The radiative forcing in Dang and Liao (2019) is the direct radiative forcing without semi-direct effects, while the other studies show total ERF values. We have added the note in Table S3.

L218-220: The factor 10 difference in ozone RF is puzzling given that the model calculating ozone changes (GEOS-Chem) is the same between the studies. I cannot understand that accounting for total column ozone change rather than tropospheric ozone change would make much of a difference (I expect changes in stratospheric ozone to be a minor contributor during this short period). I am also surprised that the uncertainty in ERF due to O3 is so large (0.81+/-0.92 W m-2 in Fig. 4b). How were the GEOS-Chem ozone data implemented in CESM2? The meteorology is different in GEOS-Chem and CESM2, so were the ozone fields implemented by cycling a single year GEOS-Chem run, as monthly mean climatologies, or in another way?

GEOS-Chem is a chemical transport model driven by reanalysis data. In Dang and Liao (2019) and this study, GEOS-Chem simulations in 2013 and 2017 were performed using MERRA-2 meteorological fields and anthropogenic emissions in 2013 and 2017, respectively. Therefore, the changes in $O_3$ between 2013 and 2017 can be attributed to the differences in both meteorology and emissions, which is easy to compare with observations. The simulated $O_3$ concentrations increased near the surface and decreased in the mid-troposphere in eastern China, leading to a net positive ERF. However, above 450 hPa, $O_3$ concentrations substantially decreased offsetting the positive ERF, which is unlikely due to the changes in anthropogenic emissions from the surface. It could be related to the interannual variability in the meteorology in higher altitudes between 2013 and 2017. To minimize this impact of the changes in meteorology, only $O_3$ data below 450 hPa from GEOS-Chem are used in CESM2 simulations, while keeping $O_3$ above 450 hPa unchanged, and are implemented by cycling the one-year data as monthly climatological mean. We have added this description in the Materials and methods section.

L220-223: Again, I do not think it makes sense to analyze surface temperature changes from these fixed SST simulations. The setup used in Zheng et al. (2020) is more logical - they used fixed SST simulations to calculate forcing and coupled atmosphere-ocean simulations to calculate surface temperature changes.

Please see our response above. We have noted the results in this study considering fast climate responses alone here.

L227-230: The current contribution from reduced emissions in the residential sector is tiny (ERF_ari of -0.03 W m-2). Can you say something about the potential for further emission reductions, i.e. how much of the BC emissions from the residential sector was reduced and how much remains?

[Figure]

Figure S4. Spatial distributions of BC emission rate from residential sector in (a) 2013 and (b) 2017, and (c) their differences (2017–2013).

We have added Fig. S4 in supplement showing BC emission rate from residential sector in 2013 and 2017 and their differences. From 2013 to 2017, only about 10% of anthropogenic BC emission from residential sector was reduced in eastern China. Previous studies have found that switching residential energy to cleaner energy prevented millions of premature deaths in China. We suggest that the use of cleaner energy in the residential sector with less BC emissions is more effective to achieve climate and health co-benefits in China in the near future. We have now discussed it in the manuscript.

L235-236: Can the authors speculate how changes in ammonium nitrate aerosols would have impacted climate?

Without the changes in ammonium nitrate data over eastern China, we cannot speculate its influences, although one study reported nitrate concentration in Beijing changed slightly related to clean air actions (Zhang et al., 2020). We have added this in the manuscript.

L252-256: It is expected that model data should also be made available, in addition to the model code.

Our model results are available at https://doi.org/10.5281/zenodo.6418003.

L446-447: Should make clear that model simulations are from CESM2 in a/c and GEOS-Chem in b/d.

We have added the description "Modelled PM$_{2.5}$ data are from CESM2 simulations in a/c and modelled O$_3$ data are from GEOS-Chem simulations in b/d." in captions of Figures 2 and 3.

L487: How is the aerosol column burden calculated, is it PM2.5 concentrations integrated over all vertical layers?

Yes, it is PM$_{2.5}$ concentrations integrated over all vertical layers.

L487-491: Several of the bars would almost disappear if the scales were not logarithmic. I think it should be made clearer by adding "Note that scales are logarithmic" or similar.

Added.

**Reference:**

Cheng, J., Tong, D., Liu, Y., Yu, S., Yan, L., Zheng, B., Geng, G., He, K., and Zhang, Q., Comparison of current and future PM2.5 air quality in China under CMIP6 and DPEC emission scenarios, Geophys. Res. Lett., 48, e2021GL093197. https://doi.org/10.1029/2021GL093197, 2021.

Dang, R. and Liao, H.: Radiative Forcing and Health Impact of Aerosols and Ozone in China as the Consequence of Clean Air Actions over 2012–2017, Geophys. Res. Lett., 46, 12511–12519, https://doi.org/10.1029/2019GL084605, 2019.

Fan, T., Liu, X., Ma, P. L., Zhang, Q., Li, Z., Jiang, Y., Zhang, F., Zhao, C., Yang, X., Wu, F., and Wang, Y.: Emission or atmospheric processes? An attempt to attribute the source of large bias of aerosols in eastern China simulated by global climate models, Atmos. Chem. Phys., 18, 1395–1417, https://doi.org/10.5194/ACP-18-1395-2018, 2018.

Fan, T., Liu, X., Wu, C., Zhang, Q., Zhao, C., Yang, X., Li, Y., Fan, T. Y., Liu, X. H., Wu, C. L., Zhang, Q., Zhao, C. F., Yang, X., and Li, Y. L.: Comparison of the Anthropogenic Emission Inventory for CMIP6 Models with a Country-Level Inventory over China and the Simulations of the Aerosol Properties, Adv. Atmos. Sci., 39, 80–96, https://doi.org/10.1007/S00376-021-1119-6, 2022.

Ghan, S. J.: Technical Note: Estimating aerosol effects on cloud radiative forcing, Atmos. Chem. Phys., 13, 9971–9974, https://doi.org/10.5194/acp-13-9971-2013, 2013.

Li, K., Jacob, D. J., Liao, H., Qiu, Y., Shen, L., Zhai, S., Bates, K. H., Sulprizio, M. P., Song, S., Lu, X., Zhang, Q., Zheng, B., Zhang, Y., Zhang, J., Lee, H. C., and Kuk, S. K.: Ozone pollution in the North China Plain spreading into the late-winter haze season, Proc. Natl. Acad. Sci. U.S.A., 118, e2015797118, https://doi.org/10.1073/pnas.2015797118, 2021.

Li, K., Jacob, D. J., Liao, H., Shen, L., Zhang, Q., and Bates, K. H.: Anthropogenic drivers of 2013–2017 trends in summer surface ozone in China, Proc. Natl. Acad. Sci. U.S.A., 116, 422–427, https://doi.org/10.1073/PNAS.1812168116, 2019a.

Li, K., Jacob, D. J., Liao, H., Zhu, J., Shah, V., Shen, L., Bates, K. H., Zhang, Q., and Zhai, S.: A two-pollutant strategy for improving ozone and particulate air quality in China, Nat. Geosci., 12, 906–910, https://doi.org/10.1038/s41561-019-0464-x, 2019b.

Li, M., Yang, Y., Wang, P., Ji, D., and Liao, H.: Impacts of strong El Niño on summertime near-surface ozone over China, Atmos. Ocean. Sci. Lett., 100193, https://doi.org/10.1016/J.AOSL.2022.100193, 2022.

Liu, L., Shawki, D., Voulgarakis, A., Kasoar, M., Samset, B. H., Myhre, G., Forster, P. M., Hodnebrog, Sillmann, J., Aalbergsjø, S. G., Boucher, O., Faluvegi, G., Iversen, T., Kirkevåg, A., Lamarque, J. F., Olivié, D., Richardson, T., Shindell, D., and Takemura, T.: A PDRMIP Multimodel Study on the Impacts of Regional Aerosol Forcings on Global and Regional Precipitation, J. Clim., 31, 4429–4447, https://doi.org/10.1175/JCLI-D-17-0439.1, 2018.

Ren, L., Yang, Y., Wang, H., Wang, P., Chen, L., Zhu, J., and Liao, H.: Aerosol transport pathways and source attribution in China during the COVID-19 outbreak, Atmos. Chem. Phys., 21, 15431–15445, https://doi.org/10.5194/acp-21-15431-2021, 2021.

Samset, B. H., Myhre, G., Forster, P. M., Hodnebrog, Andrews, T., Faluvegi, G., Fläschner, D., Kasoar, M., Kharin, V., Kirkevåg, A., Lamarque, J. F., Olivié, D., Richardson, T., Shindell, D., Shine, K. P., Takemura, T., and Voulgarakis, A.: Fast and slow precipitation responses to individual climate forcers: A PDRMIP multimodel study, Geophys. Res. Lett., 43, 2782–2791, https://doi.org/10.1002/2016GL068064, 2016.

Smith, C. J., Kramer, R. J., Myhre, G., Alterskjær, K., Collins, W., Sima, A., Boucher, O., Dufresne, J.-L., Nabat, P., Michou, M., Yukimoto, S., Cole, J., Paynter, D., Shiogama, H., O'Connor, F. M., Robertson, E., Wiltshire, A., Andrews, T., Hannay, C., Miller, R., Nazarenko, L., Kirkevåg, A., Olivié, D., Fiedler, S., Lewinschal, A., Mackallah, C., Dix, M., Pincus, R., and Forster, P. M.: Effective radiative forcing and adjustments in CMIP6 models, Atmos. Chem. Phys., 20, 9591–9618, https://doi.org/10.5194/acp-20-9591-2020, 2020.

Wang, Z., Lin, L., Xu, Y., Che, H., Zhang, X., Dong, W., Wang, C., Gui, K., and Xie, B.: Incorrect Asian aerosols affecting the attribution and projection of regional climate change in CMIP6 models, npj Clim. Atmos. Sci., 4, 2, https://doi.org/10.1038/s41612-020-00159-2, 2021.

Yang, Y., Li, M., Wang, H., Li, H., Wang, P., Li, K., Gao, M., and Liao, H.: ENSO modulation of summertime tropospheric ozone over China, Environ. Res. Lett., 17, 034020, https://doi.org/10.1088/1748-9326/ac54cd, 2022.

Yang, Y., Wang, H., Smith, S. J., Easter, R., Ma, P.-L., Qian, Y., Yu, H., Li, C., and Rasch, P. J.: Global source attribution of sulfate concentration and direct and indirect radiative forcing, Atmos. Chem. Phys., 17, 8903–8922, https://doi.org/10.5194/acp-17-8903-2017, 2017a.

Yang, Y., Wang, H., Smith, S. J., Ma, P.-L., and Rasch, P. J., Source attribution of black carbon and its direct radiative forcing in China, Atmos. Chem. Phys., 17, 4319–4336, https://doi.org/10.5194/acp-17-4319-2017, 2017b.

Yang, Y., Ren, L., Li, H., Wang, H., Wang, P., Chen, L., Yue, X., and Liao, H.: Fast Climate Responses to Aerosol Emission Reductions During the COVID-19 Pandemic, Geophys. Res. Lett., 47, e2020GL089788, https://doi.org/10.1029/2020GL089788, 2020.

Zanis, P., Akritidis, D., Georgoulias, K. A., Allen, J. R., Bauer, E. S., Boucher, O., Cole, J., Johnson, B., Deushi, M., Michou, M., Mulcahy, J., Nabat, P., Olivié, D., Oshima, N., Sima, A., Schulz, M., Takemura, T., and Tsigaridis, K.: Fast responses on pre-industrial climate from present-day aerosols in a CMIP6 multi-model study, Atmos. Chem. Phys., 20, 8381–8404, https://doi.org/10.5194/ACP-20-8381-2020, 2020.

Zeng, L., Yang, Y., Wang, H., Wang, J., Li, J., Ren, L., Li, H., Zhou, Y., Wang, P., and Liao, H.: Intensified modulation of winter aerosol pollution in China by El Niño with short duration, Atmos. Chem. Phys., 21, 10745–10761, https://doi.org/10.5194/acp-21-10745-2021, 2021.

Zhang, Z., Guan, H., Luo, L., Zheng, N., and Xiao, H.: Response of fine aerosol nitrate chemistry to Clean Air Action in winter Beijing: Insights from the oxygen isotope signatures, Sci. Total Environ., 746, 141210, https://doi.org/10.1016/J.SCITOTENV.2020.141210, 2020.

Zheng, B., Tong, D., Li, M., Liu, F., Hong, C., Geng, G., Li, H., Li, X., Peng, L., Qi, J., Yan, L., Zhang, Y., Zhao, H., Zheng, Y., He, K., and Zhang, Q.: Trends in China's anthropogenic emissions since 2010 as the consequence of clean air actions, Atmos. Chem. Phys., 18, 14095–14111, https://doi.org/10.5194/acp-18-14095-2018, 2018.

---

## Author Comment (AC2)

The manuscript by Gao et al. studies the climate responses to emission reductions in air pollutants over China due to clean air actions from 2013 to 2017, investigates both aerosols and ozone changes and their climate impacts by conducting several experiments using CESM2 model. The topic has wide implication for emission reduction policy decision making over China and fits the scope of the general ACP readership. This paper is overall well written, but there are several issues need to be addressed before the manuscript can be accepted for publication.

We thank the reviewer for the constructive suggestions, which are very helpful for improving the clarity and reliability of the manuscript. Please see our point-by-point responses (in blue) to your comments below.

**Major:**

1. The model results significantly underestimate the PM2.5 decrease compared with observation (Fig.2), which contributes to the uncertainty of this study. It would be interesting to quantify to what extent the model bias influences the estimated climate impacts.

Thank you for the suggestion. The model significantly underestimates the $PM_{2.5}$ decrease in China during 2013–2017, which is caused by many factors including strong aerosol wet removal, uncertainties in new particle formation, coarse model resolution in global climate models, the uncertainty of anthropogenic emissions of aerosols and precursor gases, the treatments of meteorology and aerosol processes, which have been reported in many previous studies (Yang et al., 2017a, b; Zeng et al., 2021; Ren et al., 2021; Fan et al., 2022, 2018). The low bias in estimated aerosol decreases may result in an underestimation of the simulated climate responses in CAM6. We have added these descriptions in the discussion section.

2. The authors investigated the climate response by conducting simulations with fixed SST at the climatological mean. I wonder how much does the slow and fast response contribute to the total climate response respectively? Though the authors stated that they will revisit this issue using a fully coupled model configuration with both fast and slow climate responses included in future studies, it is suggested to discuss the uncertainties due to neglecting the slow climate response in this paper.

Firstly, only fast climate responses are considered in our study, while the emission reductions could also influence climate response through slow oceanic processes and air-sea interactions, which can be improved by conducting fully coupled atmosphere-ocean simulations in future studies. Samset et al. (2016) showed that the fast precipitation response to changes in aerosols dominated the slow oceanic response over land of East Asia. However, to what extent the fast processes contributed to the temperature response needs further study. Neglecting the slow climate response here could lead to an incomplete aerosol climate effect.

3. I would suggest the authors to provide an in-depth discussion in the discussion section on these uncertainties, including the model bias, the neglect of slow response, the neglect of nitrate and ammonium, etc. It is better to have error bars on the simulated results or at least discuss the possible bias ranges. In addition, as stated in L215, different chemical and physical schemes contribute some uncertainties, leading to the differences compared to previous studies. Thus it is better to list the specific parameterizations of different models in Table S3.

We have substantially revised the discussion section as the following:

There are some limitations and uncertainties in the study. Firstly, only fast climate responses are considered in our study, while the emission reductions could also influence climate response through slow oceanic processes and air-sea interactions, which can be improved by conducting fully coupled atmosphere-ocean simulations in

future studies. Samset et al. (2016) showed that the fast precipitation response to changes in aerosols dominated the slow oceanic response over land of East Asia. However, to what extent the fast processes contributed to the temperature response needs further study. Neglecting the slow climate response here could lead to an incomplete aerosol climate effect. Secondly, the model significantly underestimates the $PM_{2.5}$ decrease in China during 2013–2017, which is caused by many factors including strong aerosol wet removal, uncertainties in new particle formation, the coarse model resolution, and the uncertainty in anthropogenic emissions of aerosols and precursor gases (Yang et al., 2017a, b; Zeng et al., 2021; Ren et al., 2021; Fan et al., 2022, 2018). The low bias in estimated aerosol decreases may result in an underestimation of the simulated climate responses in CAM6. Thirdly, nitrate and ammonium aerosols, which are not treated in current version of CESM2, also changed from 2013 to 2017 (Xu et al., 2019) and should have impacted on climate, although nitrate concentration in Beijing changed slightly during this time (Zhang et al., 2020). Fourthly, only 20-year simulations were performed in this study, longer simulations with ensemble members may present a more robust result. Finally, only one model is used in our study, a potential model dependence of climate responses to aerosol reductions needs further investigation using multi-model ensemble simulations.

We have also added error bars in Figure 7 and uncertainty range in Table S2.

CAM6 (CESM2) and CAM5 (CESM1) are climate models with simulation of major aerosol species, while GEOS-Chem is a chemical transport model with simulation of ozone and aerosols driven by meteorological fields from reanalysis. GEOS-Chem (http://www.geos-chem.org) is a global 3-D model of atmospheric chemistry driven by meteorological input from the Goddard Earth Observing System (GEOS). The detailed information about chemistry, aerosol process, transport, deposition, and radiation in GEOS-Chem is available at https://geos-chem.seas.harvard.edu/. CESM2/CESM1 (https://www.cesm.ucar.edu) is the coupled climate/Earth system models developed by the National Center for Atmospheric Research (NCAR). Its atmosphere model is the Community Atmosphere Model Version 6/5 (CAM6/CAM5). The detail information about chemical and physical schemes and the changes between CAM5 and CAM6 are available in Danabasoglu et al. (2020). We have added these descriptions in Table S3.

**Minor:**

1. L79, 'A comprehensive consideration of aerosol/O3-radiation and aerosol-cloud interactions are included in the model.' How are these processes considered specifically in the model? I suggest authors to introduce these schemes in detail, or at least show some references.

In CESM2-CAM6, aerosols are treated using the Modal Aerosol Model version 4 (MAM4; Liu et al., 2016). The Morrison-Gettelman cloud microphysics scheme version 2 (MG2, Gettelman and Morrison, 2015) is applied to forecast mass and number concentrations of rain and snow. The mixed phase ice nucleation depending on aerosols is also included (Hoose et al., 2010; Wang et al., 2014). Radiation transfer scheme uses Rapid Radiative Transfer Model for General circulation models (RRTMG, Iacono et al., 2008). Ozone mixing ratio is prescribed for use in radiative transfer calculations. We have added this information in the manuscript.

2. L83, it is better to list some reference about ozone simulation in GEOS-Chem here.

We have added more information and references for GEOS-Chem simulation as "Global three-dimensional tropospheric monthly $O_3$ concentrations below 450 hPa for years 2013 and 2017 are adopted from simulations using GEOS-Chem model v12.9.3, considering that it has a good performance in simulating ozone concentration changes during 2013–2017 (Li et al., 2019a, b, 2021). GEOS-Chem is a global model of atmospheric chemistry with fully coupled $O_3$–$NO_x$–hydrocarbon–aerosol chemical mechanisms, which has a horizontal resolution of

2° latitude × 2.5° longitude and 47 vertical layers driven by the MERRA-2 (Modern-Era Retrospective analysis for Research and Applications Version 2) meteorological fields. The model simulations in 2013 and 2017 with one-year spin up use the same aerosol and precursor gas emissions as used in CAM6 and the results are interpolated to the same resolution used in CAM6. The details of the GEOS-Chem model simulations can be found in Li et al. (2022) and Yang et al. (2022). Note that, GEOS-Chem model presents a strong decrease in $O_3$ concentrations in upper troposphere between 2013 and 2017, which is mainly attributed to the varying meteorological fields between 2013 and 2017. To minimize the impacts from the changes in meteorology, only $O_3$ data below 450 hPa from GEOS-Chem are used in CESM2 simulations, while keeping $O_3$ above 450 hPa unchanged, and are implemented by cycling the one-year data as monthly climatological mean."

3. In section 2, please add some introductions about observations used in this study.

We have added the sentence: "Hourly observations of $PM_{2.5}$ and $O_3$ concentrations across China in 2013 and 2017 derived from the China National Environmental Monitoring Centre (CNEMC) are applied to evaluate the model performance." in section 2.

4. L133, change 'other sub-regions' to 'over other sub-regions'.

Changed.

5. L136-L137, better to list some references here.

Added the reference (Li et al., 2019).

6. Figure 1, better to mention the MEIC inventory in figure caption.

We have added the sentence: "The anthropogenic emission data are derived from MEIC." in figure 1 caption.

7. Figure S1, the color bar is not shown.

Revised.

**Reference:**

Danabasoglu, G., Lamarque, J. F., Bacmeister, J., Bailey, D. A., DuVivier, A. K., Edwards, J., Emmons, L. K., Fasullo, J., Garcia, R., Gettelman, A., Hannay, C., Holland, M. M., Large, W. G., Lauritzen, P. H., Lawrence, D. M., Lenaerts, J. T. M., Lindsay, K., Lipscomb, W. H., Mills, M. J., Neale, R., Oleson, K. W., Otto-Bliesner, B., Phillips, A. S., Sacks, W., Tilmes, S., van Kampenhout, L., Vertenstein, M., Bertini, A., Dennis, J., Deser, C., Fischer, C., Fox-Kemper, B., Kay, J. E., Kinnison, D., Kushner, P. J., Larson, V. E., Long, M. C., Mickelson, S., Moore, J. K., Nienhouse, E., Polvani, L., Rasch, P. J., and Strand, W. G.: The Community Earth System Model Version 2 (CESM2), J. Adv. Model. Earth Syst., 12, e2019MS001916, https://doi.org/10.1029/2019MS001916, 2020.

Fan, T., Liu, X., Ma, P. L., Zhang, Q., Li, Z., Jiang, Y., Zhang, F., Zhao, C., Yang, X., Wu, F., and Wang, Y.: Emission or atmospheric processes? An attempt to attribute the source of large bias of aerosols in eastern China

simulated by global climate models, Atmos. Chem. Phys., 18, 1395–1417, https://doi.org/10.5194/ACP-18-1395-2018, 2018.

Fan, T., Liu, X., Wu, C., Zhang, Q., Zhao, C., Yang, X., Li, Y., Fan, T. Y., Liu, X. H., Wu, C. L., Zhang, Q., Zhao, C. F., Yang, X., and Li, Y. L.: Comparison of the Anthropogenic Emission Inventory for CMIP6 Models with a Country-Level Inventory over China and the Simulations of the Aerosol Properties, Adv. Atmos. Sci., 39, 80–96, https://doi.org/10.1007/S00376-021-1119-6, 2022.

Gettelman, A., and Morrison, H.: Advanced two-moment bulk microphysics for global models. Part I: Off-line tests and comparison with other schemes, J. Clim., 28, 1268–1287, https://doi.org/10.1175/JCLI-D-14-00102.1, 2015.

Hoose, C., Kristjánsson, J. E., Chen, J.-P., and Hazra, A.: A classical-theory-based parameterization of heterogeneous ice nucleation by mineral dust, soot, and biological particles in a global climate model, J. Atmos. Sci., 67, 2483–2503, https://doi.org/ 10.1175/2010JAS3425.1, 2010.

Iacono, M. J., Delamere, J. S., Mlawer, E. J., Shephard, M. W., Clough, S. A., and Collins, W. D.: Radiative forcing by long-lived greenhouse gases: Calculations with the AER radiative transfer models, J. Geophys. Res. Atmos., 113, D13103, https://doi.org/10.1029/2008JD009944, 2008.

Li, K., Jacob, D. J., Liao, H., Qiu, Y., Shen, L., Zhai, S., Bates, K. H., Sulprizio, M. P., Song, S., Lu, X., Zhang, Q., Zheng, B., Zhang, Y., Zhang, J., Lee, H. C., and Kuk, S. K.: Ozone pollution in the North China Plain spreading into the late-winter haze season, Proc. Natl. Acad. Sci. U.S.A., 118, e2015797118, https://doi.org/10.1073/pnas.2015797118, 2021.

Li, K., Jacob, D. J., Liao, H., Shen, L., Zhang, Q., and Bates, K. H.: Anthropogenic drivers of 2013–2017 trends in summer surface ozone in China, Proc. Natl. Acad. Sci. U.S.A., 116, 422–427, https://doi.org/10.1073/PNAS.1812168116, 2019a.

Li, K., Jacob, D. J., Liao, H., Zhu, J., Shah, V., Shen, L., Bates, K. H., Zhang, Q., and Zhai, S.: A two-pollutant strategy for improving ozone and particulate air quality in China, Nat. Geosci., 12, 906–910, https://doi.org/10.1038/s41561-019-0464-x, 2019b.

Li, M., Yang, Y., Wang, P., Ji, D., and Liao, H.: Impacts of strong El Niño on summertime near-surface ozone over China, Atmos. Ocean. Sci. Lett., 100193, https://doi.org/10.1016/J.AOSL.2022.100193, 2022.

Liu, X., Ma, P.-L., Wang, H., Tilmes, S., Singh, B., Easter, R. C., Ghan, S. J., and Rasch, P. J.: Description and evaluation of a new four-mode version of the Modal Aerosol Module (MAM4) within version 5.3 of the Community Atmosphere Model, Geosci. Model Dev., 9, 505–522, https://doi.org/10.5194/gmd-9-505-2016, 2016.

Ren, L., Yang, Y., Wang, H., Wang, P., Chen, L., Zhu, J., and Liao, H.: Aerosol transport pathways and source attribution in China during the COVID-19 outbreak, Atmos. Chem. Phys., 21, 15431–15445, https://doi.org/10.5194/acp-21-15431-2021, 2021.

Samset, B. H., Myhre, G., Forster, P. M., Hodnebrog, Andrews, T., Faluvegi, G., Fläschner, D., Kasoar, M., Kharin, V., Kirkevåg, A., Lamarque, J. F., Olivié, D., Richardson, T., Shindell, D., Shine, K. P., Takemura, T., and Voulgarakis, A.: Fast and slow precipitation responses to individual climate forcers: A PDRMIP multimodel study, Geophys. Res. Lett., 43, 2782–2791, https://doi.org/10.1002/2016GL068064, 2016.

Wang, Y., Liu, X., Hoose, C., and Wang, B.: Different contact angle distributions for heterogeneous ice nucleation in the Community Atmospheric Model version 5. Atmos. Chem. Phys., 14, 10411–10430, https://doi.org/10.5194/acpd-14-10411-2014, 2014.

Xu, Q., Wang, S., Jiang, J., Bhattarai, N., Li, X., Chang, X., Qiu, X., Zheng, M., Hua, Y. and Hao, J.: Nitrate dominates the chemical composition of PM2.5 during haze event in Beijing, China, Sci. Total Environ., 689, 1293–1303, https://doi.org/10.1016/j.scitotenv.2019.06.294, 2019.

Yang, Y., Li, M., Wang, H., Li, H., Wang, P., Li, K., Gao, M., and Liao, H., ENSO modulation of summertime tropospheric ozone over China, Environ. Res. Lett., 17, 034020, https://doi.org/10.1088/1748-9326/ac54cd, 2022.

Yang, Y., Wang, H., Smith, S. J., Easter, R., Ma, P.-L., Qian, Y., Yu, H., Li, C., and Rasch, P. J.: Global source attribution of sulfate concentration and direct and indirect radiative forcing, Atmos. Chem. Phys., 17, 8903–8922, https://doi.org/10.5194/acp-17-8903-2017, 2017a.

Yang, Y., Wang, H., Smith, S. J., Ma, P.-L., and Rasch, P. J., Source attribution of black carbon and its direct radiative forcing in China, Atmos. Chem. Phys., 17, 4319–4336, https://doi.org/10.5194/acp-17-4319-2017, 2017b.

Zeng, L., Yang, Y., Wang, H., Wang, J., Li, J., Ren, L., Li, H., Zhou, Y., Wang, P., and Liao, H.: Intensified modulation of winter aerosol pollution in China by El Niño with short duration, Atmos. Chem. Phys., 21, 10745–10761, https://doi.org/10.5194/acp-21-10745-2021, 2021.

Zhang, Z., Guan, H., Luo, L., Zheng, N., and Xiao, H.: Response of fine aerosol nitrate chemistry to Clean Air Action in winter Beijing: Insights from the oxygen isotope signatures, Sci. Total Environ., 746, 141210, https://doi.org/10.1016/J.SCITOTENV.2020.141210, 2020.